# Continuous-Time Hidden Markov Factor Model for Mobile Health Data: Application to Adverse Posttraumatic Neuropsychiatric Sequelae

Lin Ge, Xinming An*, Donglin Zeng, Samuel McLean, Ronald Kessler, and Rui Song*

*Abstract*—Adverse posttraumatic neuropsychiatric sequelae (APNS) are common among veterans and millions of Americans after traumatic exposures, resulting in substantial health and financial burdens for trauma survivors, their families, and society. Despite numerous studies conducted on APNS over the past decades, there has been limited progress in understanding the underlying neurobiological mechanisms due to several unique challenges. One of these challenges is the reliance on subjective self-report measures to assess APNS, which can easily result in measurement errors and biases (e.g., recall bias). To mitigate this issue, in this paper, we investigate the potential of leveraging objective longitudinal mobile device data to identify homogeneous APNS states and study the dynamic transitions among them and potential risk factors after trauma exposure. To handle the unique challenges posed by longitudinal mobile device data, we developed a continuous-time hidden Markov factor model and designed a Stabilized Expectation-Maximization algorithm for parameter estimation. Simulation studies were conducted to evaluate the performance of parameter estimation and model selection. Finally, to demonstrate the practical utility of the method, we applied it to mobile device data collected from the Advancing Understanding of RecOvery afteR traumA (AURORA) study. A Python implementation of the proposed method is available at https://anonymous.4open.science/r/CTHMFM.

*Index Terms*—Continuous-time hidden Markov model, Mental health, Multivariate longitudinal data

## I. INTRODUCTION

Adverse posttraumatic neuropsychiatric sequelae (APNS) (e.g., pain, depression, and PTSD) are frequently observed in civilians and military veterans who have experienced traumatic events. These APNS increase the risk of chronic illnesses, including cancer and heart disease, and substantially contribute to drug abuse, suicide, and disability. Moreover, APNS impose enduring psychosocial and financial burdens not only on individuals with the disorder but also on their families, communities, and society as a whole. However, little progress

The research is partially supported by NSF under Grant DMS-1555244 and 2113637, NIMH under Grant U01MH110925, the US Army Medical Research and Material Command, The One Mind Foundation, and The Mayday Fund. Lin Ge is with the Department of Statistics, North Carolina State University, Raleigh, NC 27695 USA (e-mail: lge@ncsu.edu). Xinming An is with the Department of Anesthesiology, The University of North Carolina at Chapel Hill, Chapel Hill, NC 27514 USA (e-mail: Xinming_An@med.unc.edu). Donglin Zeng is with the Department of Biostatistics, The University of North Carolina at Chapel Hill, Chapel Hill, NC 27599 USA (e-mail: dzeng@bios.unc.edu). Samuel McLean is with the Department of Psychiatry, The University of North Carolina at Chapel Hill, Chapel Hill, NC 27514 USA (e-mail: samuel_mclean@med.unc.edu). Ronald Kessler is with the Department of Health Care Policy, Harvard Medical School, Boston, MA 02115 USA (e-mail: kessler@hcp.med.harvard.edu). Rui Song is with the Department of Statistics, North Carolina State University, Raleigh, NC 27695 USA (e-mail: rsong@ncsu.edu). *Drs. Rui Song and Xinming An made equal contributions to this manuscript and serve as co-corresponding authors.

has been made in advancing APNS research over the past few decades due to unique challenges. First, APNS have been evaluated through subjective self-reported measures, which lack objective reliability. Second, the heterogeneity among patients, as recognized in traditional classification and diagnoses, complicates the study of APNS. Lastly, these APNS disorders are often studied and treated independently, despite their frequent co-occurrence [1]. These obstacles hinder the identification of objective markers, the advancement in understanding the neurobiological mechanisms of APNS, and the development of effective preventative/treatment strategies.

Identifying homogeneous states and exploring the dynamic prognosis of APNS in the immediate aftermath of trauma exposure holds promise for enhancing our understanding of APNS and identifying effective intervention options and appropriate timing at the individual level. Regrettably, due to the lack of appropriate data and effective statistical method, no large-scale studies have been conducted to investigate the onset, dynamic transitions (such as recovery and relapse), and associated risk factors of APNS. To help address the challenges, the National Institutes of Mental Health, joined by the US Army Medical Research and Material Command, developed the Advancing Understanding of RecOvery afteR traumA (AURORA) study [1]. This study gathered biobehavioral data from a large cohort of trauma survivors (n = 2,997) across the United States over a year, including self-reported surveys, web-based neurocognitive tests, digital phenotyping data (i.e., from wrist wearables and smartphones), psychophysical tests, neuroimaging assessments, and genomics data. In contrast to previous studies relying on self-report surveys or neuroimages [2], our work aims to address the challenges of APNS by utilizing the objective digital phenotyping data that tracks individuals' behaviors, moods, and health statuses in real-time, real-life environments. Specifically, we developed a Hidden Markov Factor Model (HMFM) to analyze the mobile device data, allowing us to simultaneously identify homogeneous subtypes, investigate subtype-specific structures, and model individual progression with associated risk factors.

Hidden Markov Models (HMMs) have been widely used in various fields [3]. However, mobile device data presents two unique challenges that standard HMMs cannot handle, including the interdependent variables with unknown interrelationship structures and unevenly spaced measurements.

Mobile device sensor data, such as accelerometer and photoplethysmography (PPG) from smartwatches, involves highly

intensive time series data. Instead of being directly used, these data are typically pre-processed to generate technical summaries that represent various characteristics of each time series variable, which are often highly correlated. As the number of features increases, the parameters in the covariance matrix expand exponentially, making a fully free covariance matrix impractical. Consequently, when implementing HMMs, features are usually assumed to be independent given the latent state membership, despite their correlation, potentially introducing estimation bias.

To appropriately model the association between features, factor analysis models (FMs) [4] provide an efficient and parsimonious approach and have been incorporated into HMMs in various ways. For example, the factor analyzed hidden Markov model [5] combines an FM with a discrete-time HMM (DTHMM), which assumes evenly spaced measurements. It has been extensively used in a variety of real-world applications, including speech recognition [6], environmental protection [7], and seizure detection [8]. Similarly, [9] introduced the regime-switching factor model to handle high-dimensional financial market data. However, they all assume homogeneous transition probability matrices, limiting their ability to account for the heterogeneity of transition probabilities over time and among different subjects and explore risk factors of state transition. To simultaneously capture the interrelationships among observed features and account for the variability of transition probabilities, a joint framework incorporating HMM, FM, and a feature-based transition model was recently proposed [10], [11]. However, it is not directly applicable to mobile device data. First, it employs a confirmatory factor model (CFM) with pre-specified structures for the factor loading matrices, which are often unknown a priori. Therefore, an exploratory factor model (EFM) is needed to explore the interrelationships among all observed features. Second, their framework assumes ordered states, which is inappropriate for our use case.

Another challenge posed by mobile device data is the irregular spacing of measurements. For example, activity and heart rate variability (HRV) data were collected only when the participants wore the watches, resulting in non-uniformly spaced observations and significant variation in sampling schedules between individuals. While the aforementioned methods are all based on DTHMM, neglecting the impact of time gaps between consecutive observations on transition rates, continuous-time discrete-state HMM (CTHMM) was developed to handle irregularly spaced measurements [12]. CTHMM and its extensions that incorporate covariates to characterize transition rates are widely used in medical research [13], [14]. However, none of them address the interrelationships among features.

In this paper, to simultaneously address the two challenges and examine heterogeneous transition patterns, we propose to use Continuous-Time HMFM (CT-HMFM), integrating CTHMM, EFM, and a feature-based transition model. Our contributions are as follows: First, we examine the utility of data collected in an open environment from consumer-grade mobile devices for mental health research. This contrasts with most existing studies on data collected in controlled lab environments. Second, we propose CT-HMFM to address the unique challenges introduced by mobile device data and depict the non-homogeneous state transition processes of multiple individuals. Simulation studies using synthetic data demonstrate exceptional parameter estimation and model selection performance. Finally, we analyze HRV and activity data from the AURORA study, followed by interpretations and discussions of biological findings that highlight the immense potential of mobile health data and our proposed method for mental health research.

## II. AURORA DATASET

In the AURORA study [1], data were collected from multiple sources, with our analysis focusing on the accelerometer and PPG data collected by the Verily smartwatches. Given the known associations with APNS [15], [16], the accelerometer data were used to quantify physical activity features, while the PPG data were pre-processed to extract heart rate variability (HRV) metrics (for details on the raw data and pre-processing steps, refer to [17]). Briefly, activity features are extracted during a 24-hour window to evaluate daily activity patterns. After converting accelerometer data to activity counts [18], *meanAcc* is the average activity counts. *Amplitude*, a cosinor rhythmometry feature, is computed to capture circadian rhythms [19]. Using the Cole-Kripke algorithm [20], accelerometry epochs are classified into wake or sleep states, and the *SWCK* quantifies transition rates between wake and sleep. Additionally, the average activity during the five least active hours (*L5*) is calculated from raw accelerometer data [21], representing nighttime activity.

HRV features were derived from PPG data by first calculating and denoising beat-to-beat (BB) interval [22] time series to obtain normal-to-normal (NN) intervals, which were then analyzed using a 5-minute sliding window. Selected HRV features for this study include average heart rate (*NNmean*), skewness (*NNskew*), and standard deviation (*SDNN*) of NN intervals, indicating heart rate variability and rapid changes. *Lfhf* is the low-frequency to high-frequency power ratio, with extremes suggesting parasympathetic or sympathetic dominance [22]. *DC* serves as a mortality risk indicator in cardiac conditions [23]. The lower the *DC* index, the higher the mortality risk. Additionally, *SD1SD2* and *ApEn* are used to quantify the unpredictability and regularity of successive heartbeats (R-R interval). To match the activity data, daily statistical summaries of each HRV feature are used, including mean, minimum, maximum, interquartile range, and variance.

In this study, we focused on survivors of motor vehicle collision trauma. Since the data collection depends on the participants' wearing of the devices, missing data is a common issue. To assure data quality, we include only records that have complete activity data and a positive wake percentage. For HRV data, ideally, an individual can have 2,880 records per day. We retain only those days where at least 30% of these records are available to ensure that our daily summary statistics are representative. The final dataset consists of daily summaries of activity and HRV features from 258 patients, with each providing at least 50 days of records.

## III. CT-HMFM

Motivated by the structures of the processed AURORA datasets, we consider data in the form of repeated measurements of $p$ features over $T_i$ occasions for each individual $i$ of $N$ subjects. The proposed models are in the framework of HMM. Let $w_{it}$ be the latent state of individual $i$ at occasion $t$, taking value from the finite discrete set $\{1, \cdots, J\}$. $\boldsymbol{W}_i = (w_{i1}, \cdots, w_{iT_i})$ is the state sequence over $T_i$ repeated measurements. Let $\boldsymbol{P}_{it}$ be the $J \times J$ transition probability matrix for individual $i$ at occasion $t$, $t = \{2, \cdots, T_i\}$, of which the $(k, j)$ entry is $\boldsymbol{P}_{it,kj} = P(w_{it} = j|w_{i,t-1} = k)$, and $\boldsymbol{P}_{it,kk} = 1 - \sum_{j:j\neq k} \boldsymbol{P}_{it,kj}$. At $t = 1$, we assume that the initial state follows a multinomial distribution with probabilities $\boldsymbol{\pi} = (\pi_1, \cdots, \pi_J)'$, such that $\sum_{i=1}^{J} \pi_i = 1$. The objective of the HMM is to delineate latent Markov processes given observations by estimating the transition probability matrix $\boldsymbol{P}$ and the initial state distribution. Distinct from traditional HMMs, our model further incorporates two additional components to meet our objectives. The first component, detailed in Section III-A, is a state-specific measurement model using EFM to uncover interrelationships among variables. The second component, outlined in Section III-B, is a transition model (TM) to capture heterogeneous transition patterns.

### A. State-Specific Measurement Model

Let $\boldsymbol{y}_{it}$ denote a vector of the observed value of $p$ features for subject $i$ at time $t$. $\boldsymbol{z}_{it}$ is a $K$ dimensional vector of latent scores assumed to be independent of $w_{it}$ and following a standard multivariate normal distribution. While we assume that $K$ is constant across states, our model can easily be extended to accommodate varying $K_j$. For each individual $i$, $\boldsymbol{Y}_i = (\boldsymbol{y}_{i1}, \cdots, \boldsymbol{y}_{iT_i})$ is a $p \times T_i$ matrix containing all measurements and $\boldsymbol{Z}_i = (\boldsymbol{z}_{i1}, \cdots, \boldsymbol{z}_{iT})$ is a $K \times T_i$ matrix containing all latent features. The first component of our model is an FM, with the primary goal of identifying the interrelationship structures between observed response variables and the underlying constructions of latent variables. For individual $i$ at time $t$, given $w_{it} = j$, the FM assumes that:

$$[\boldsymbol{y}_{it}|w_{it} = j] = \boldsymbol{\mu}_j + \boldsymbol{\Lambda}_j \boldsymbol{z}_{it} + \boldsymbol{e}_{it},$$
$$\boldsymbol{z}_{it} \overset{i.i.d.}{\sim} \mathcal{N}(\boldsymbol{0}, \boldsymbol{I}_K), \boldsymbol{e}_{it} \overset{i.i.d.}{\sim} \mathcal{N}(\boldsymbol{0}, \boldsymbol{\Psi}), \boldsymbol{z}_{it} \perp\!\!\!\perp \boldsymbol{e}_{it}, \quad (1)$$

where $\boldsymbol{\mu}_j$ is a $p \times 1$ vector of state-specific expected mean response, $\boldsymbol{\Lambda}_j$ is a $p \times K$ state-specific factor loading matrix, $\boldsymbol{\Psi}$ is a $p \times p$ diagonal covariance matrix for the error term $\boldsymbol{e}_{it}$ with positive nonconstant diagonal entries. Alternatively, (1) can be expressed as $[\boldsymbol{y}_{it}|w_{it} = j] \overset{i.i.d.}{\sim} \mathcal{N}(\boldsymbol{\mu}_j, \boldsymbol{\Lambda}_j \boldsymbol{\Lambda}_j' + \boldsymbol{\Psi})$. It is crucial to emphasize that, unlike CFM with pre-specified structures of factor loading matrices, our approach imposes no assumptions on $\boldsymbol{\Lambda}_j$. Therefore, the structure of $\boldsymbol{\Lambda}_j$ is entirely data-driven, making the first component of (1) an EFM.

### B. Transition Model

Given a state sequence $\boldsymbol{W}_i$, standard assumptions of HMM assume that 1) given a state $w_{it}$, observations $\boldsymbol{y}_{it}$ are independent, and 2) given a state $w_{it}$ and subjects' contextual features, the state at the subsequent occasion $w_{i,t+1}$ is unrelated to any information from previous occasions.

Taking into account the effects of the time interval, the continuous-time Markov process relies not on the transition probability matrix $\boldsymbol{P}$, but on a transition intensity matrix $\boldsymbol{Q}$ [24]. This matrix $\boldsymbol{Q}$ is the limit of $\boldsymbol{P}$ as the time interval approaches zero. Suppose that $\delta_{it}$ is the number of pre-specified time units between $t^{th}$ and $(t-1)^{th}$ observation, then the transition intensity for subject $i$ from state $j$ to state $k$ at time $t$ is

$$q_{jk} = \lim_{\delta_{it} \to 0} \frac{P(w_{it} = k|w_{i,t-1} = j)}{\delta_{it}} > 0, j \neq k,$$

and $q_{jj} = -\sum_{k \neq j} q_{jk}$. The corresponding transition probability matrix $\boldsymbol{P}(\delta_{it})$ can be calculated as the matrix exponential of $\delta_{it} * \boldsymbol{Q}$. The time intervals are assumed to be independent.

To investigate the impact of covariates on transition rates, the transition intensity matrix can be modeled through a log-linear model [25], such that $\log(q_{jk}|\boldsymbol{x}_{it}) = \boldsymbol{x}_{it}' \boldsymbol{B}_{jk}$, where $\boldsymbol{x}_{it}$ is a $d \times 1$ vector of vector of covariates for individual $i$ at time $t$, and $\boldsymbol{B}_{jk}$ is a state-specific $d \times 1$ vector of fixed effects coefficients. The $\boldsymbol{B}_{jk}$ intends to quantify the effect of covariates on the probability of transitioning from state $j$ to a different state $k$ to provide an understanding of how covariates influence transition patterns and investigate the potential risk factors. Calculating the exponential of a matrix can be challenging. For computational efficiency, we approximate the $\exp(\boldsymbol{Q})$ using the $(\boldsymbol{I} + \boldsymbol{Q}/a)^a$ for some sufficiently large $a$ [26].

## IV. EXPECTATION-MAXIMIZATION ALGORITHM

Let $\boldsymbol{\lambda} = (\{\boldsymbol{\mu}_j\}_{j=1}^{J}, \{\boldsymbol{\Lambda}_j\}_{j=1}^{J}, \boldsymbol{\Psi}, \{\boldsymbol{B}_{kj}\}_{k,j=1}^{J}, \boldsymbol{\pi})$. Given the sequence of latent states $\boldsymbol{W}_i$ and the latent factor scores $\boldsymbol{Z}_i$ for each $i$, a joint probability distribution of the observations and all latent variables, $L_{ci}(\boldsymbol{\lambda})$, can be constructed as follows:

$$P(w_{i1}) \times \prod_{t=2}^{T_i} P(w_{it}|w_{i,t-1}, \boldsymbol{x}_{it}) \times \prod_{t=1}^{T_i} P(\boldsymbol{y}_{it}|w_{it}, \boldsymbol{z}_{it})P(\boldsymbol{z}_{it}). \tag{2}$$

By the independence property of $\boldsymbol{Y}_i$, $\boldsymbol{W}_i$, and $\boldsymbol{Z}_i$ across $i$, the complete likelihood function $(L_c)$ for the whole sample can be obtained by taking the product of (2) over $i$.

Our goal is to estimate $\boldsymbol{\lambda}$ by maximizing the likelihood function $L_c$, or its logarithm $l_c$. Since both $\boldsymbol{W}_i$ and $\boldsymbol{Z}_i$ are unobserved, the expectation-maximization (EM) algorithm is commonly used to identify the maximum likelihood estimator. As the name suggests, the EM algorithm finds a local maximum of the marginal likelihood by iteratively applying the expectation and maximization steps discussed below.

### A. Expectation Step (E-step)

The E-step gets the expectation of $l_c$ given observations, with respect to the current conditional distribution of unobserved variables and the current parameter estimates $\boldsymbol{\lambda}^v$. Denote the target expectation (i.e., $E_{\boldsymbol{\lambda}^v}[l_c(\boldsymbol{\lambda})|\boldsymbol{Y}, \boldsymbol{X}]$) as $\Omega(\boldsymbol{\lambda}, \boldsymbol{\lambda}^v)$. While an explicit form of the probability density function of $z_{it}$ exists, the calculation of conditional state probabilities can be computationally heavy. Therefore, we utilize a scaled version of the forward-backward algorithm (FBA) [27] to get the conditional state probabilities efficiently.

Specifically, we first define the forward probability $\alpha_{ij}(t)$ as $P(w_{it} = j | \boldsymbol{y}_{i1}, \cdots, \boldsymbol{y}_{it})$. Denote $P_j(\boldsymbol{y}_{it})$ the probability density function of $\boldsymbol{y}_{it}$ given $w_{it} = j$ and $c_i(t)$ the conditional probability of observation $\boldsymbol{y}_{it}$ given all past observations. Note that we omit the dependence of probabilities on $\delta_{it}$ in notations for brevity. For each individual $i$ and state $j$, using a recursion scheme, the forward probabilities at $t = 1, \cdots, T_i$ will be calculated as:

$$\alpha_{ij}(1) = \frac{\pi_j P_j(\boldsymbol{y}_{i1})}{\sum_{j=1}^J \pi_j P_j(\boldsymbol{y}_{i1})} = \frac{\pi_j P_j(\boldsymbol{y}_{i1})}{c_i(1)};$$

$$\alpha_{ij}(t) = \frac{P_j(\boldsymbol{y}_{it})[\sum_{k=1}^J \alpha_{ik}(t-1)\boldsymbol{P}_{itkj}]}{c_i(t)},$$

where $c_i(t) = \sum_{j=1}^J P_j(\boldsymbol{y}_{it})[\sum_{k=1}^J \alpha_{ik}(t-1)\boldsymbol{P}_{itkj}]$. Then, we define the backward probability $\beta_{ij}(t)$ as $\frac{P(\boldsymbol{y}_{i,t+1}, \cdots, \boldsymbol{y}_{i,T_i} | w_{it}=j, \boldsymbol{\lambda})}{c_i(t+1)}$. Similarly, we define a recursion form to update the backward probabilities at $t = T_i, \cdots, 1$:

$$\beta_{ij}(T_i) = 1; \beta_{ij}(t) = \frac{\sum_{k=1}^J \boldsymbol{P}_{i,t+1,jk} P_k(\boldsymbol{y}_{i,t+1})\beta_{ik}(t+1)}{c_i(t+1)}.$$

After that, in the smoothing step, denote $\epsilon_{ikj}^v(t)$ as $P(w_{i,t} = j, w_{i,t-1} = k | \boldsymbol{Y}_i, \boldsymbol{\lambda}^v)$ and $\gamma_{ij}^v(t)$ as $P(w_{it} = j | \boldsymbol{Y}_i, \boldsymbol{\lambda}^v)$. The target conditional state probabilities are functions of the forward probability and backward probability as follows:

$$\gamma_{ij}^v(t) = \alpha_{ij}(t)\beta_{ij}(t); \epsilon_{ikj}^v(t) = \frac{\alpha_{ik}^v(t-1)\boldsymbol{P}_{itkj}P_j(\boldsymbol{y}_{it})\beta_{ij}^v(t)}{c_i(t)}.$$

Then, the $\Omega(\boldsymbol{\lambda}, \boldsymbol{\lambda}^v)$ can be written as

$$constant + h(\boldsymbol{\pi}) + h(\{\boldsymbol{B}_{kj}\}_{k,j=1}^J) - \frac{1}{2}h(\boldsymbol{\Psi}, \{\boldsymbol{\Lambda}_j\}_{j=1}^J, \{\boldsymbol{\mu}_j\}_{j=1}^J),$$

where $h(\boldsymbol{\pi})$ depends on the initial state distribution, $h(\{\boldsymbol{B}_{kj}\}_{k,j=1}^J)$ depends on the probability transition matrix, and $h(\boldsymbol{\Psi}, \{\boldsymbol{\Lambda}_j\}_{j=1}^J, \{\boldsymbol{\mu}_j\}_{j=1}^J)$ is a function of parameters $\boldsymbol{\Psi}$, $\boldsymbol{\Lambda}_j$, and $\boldsymbol{\mu}_j$. Explicit forms are provided in Appendix A-A.

### B. Maximization (M-step)

Within each M-step, since $h(\boldsymbol{\Psi}, \{\boldsymbol{\Lambda}_j\}, \{\boldsymbol{\mu}_j\})$, $h(\boldsymbol{\pi})$, and $h(\{\boldsymbol{B}_{kj}\})$ do not share parameters, we maximize each of them separately. The estimator of $\boldsymbol{\pi}$, $\boldsymbol{\Lambda}_j$, $\boldsymbol{\mu}_j$, and $\boldsymbol{\Psi}$ can be directly derived by setting $h(\boldsymbol{\pi}) = 0$ and $h(\boldsymbol{\Psi}, \{\boldsymbol{\Lambda}_j\}, \{\boldsymbol{\mu}_j\}) = 0$ (see Appendix A-B for details). For $\{\boldsymbol{B}_{kj}\}_{k,j=1}^J$, a one-step Fisher scoring (FS) [28] is implemented.

Let $\boldsymbol{\theta}$ be a vector of all transition model parameters, such that $\boldsymbol{\theta} = vec(\{\boldsymbol{B}_{kj}'\}, k \neq j)$. Recalling the first derivative of matrix exponential [14] and using Theorem 1 in [29],

$$\frac{\partial}{\partial \theta_u} exp(\boldsymbol{A}(\theta_u)) = exp(\begin{bmatrix} \boldsymbol{A}(\theta_u) & \tilde{\boldsymbol{A}}(\theta_u) \\ \boldsymbol{0} & \boldsymbol{A}(\theta_u) \end{bmatrix})_{0:J, J:2J},$$

where $\tilde{\boldsymbol{A}}(\theta_u) = (\tilde{A}_{ij}(\theta_u)) = (\frac{\partial A_{ij}(\theta_u)}{\partial \theta_u})$. Denote $\frac{\partial \boldsymbol{P}_{kj}(\delta_{it})}{\partial \theta_u}$ as the $(k, j)$ entry of the first derivative of $\boldsymbol{P}(\delta_{it})$ with respect to $\theta_u$ (i.e., the $u^{th}$ entry of $\boldsymbol{\theta}$). Having the first derivative of $\boldsymbol{P}(\delta_{it}) = exp(\delta_{it} * \boldsymbol{Q})$ with respect to each component of $\boldsymbol{\theta}$ calculated accordingly, the FS can be directly implemented to update $\boldsymbol{\theta}$ to forbid the calculation of the second derivative

of matrix exponential. Specifically, denote $\boldsymbol{S}^*$ be the score function,

$$\boldsymbol{S}_u^*(\boldsymbol{\theta}) = \sum_{i=1}^N \sum_{t=2}^{T_i} \sum_{j=1}^J \sum_{k=1}^J \frac{\epsilon_{ikj}^v(t)}{\boldsymbol{P}_{kj}(\delta_{it})} \frac{\partial \boldsymbol{P}_{kj}(\delta_{it})}{\partial \theta_u}.$$

Let $\boldsymbol{M}^*$ be the negative Fisher information matrix. Its $(u, v)$ entry $\boldsymbol{M}_{uv}^*$ is in the form of:

$$\boldsymbol{M}_{uv}^*(\boldsymbol{\theta}) = \sum_{i=1}^N \sum_{t=2}^{T_i} \sum_{j=1}^J \sum_{k=1}^J \frac{\gamma_{ik}^v(t-1)}{\boldsymbol{P}_{kj}(\delta_{it})} \frac{\partial \boldsymbol{P}_{kj}(\delta_{it})}{\partial \theta_u} \frac{\partial \boldsymbol{P}_{kj}(\delta_{it})}{\partial \theta_v}.$$

After getting both the score function and the Fisher information matrix, parameters $\boldsymbol{\theta}$ can be updated as $\boldsymbol{\theta}^{v+1} = \boldsymbol{\theta}^v + \boldsymbol{M}^*(\boldsymbol{\theta}^v)^{-1}\boldsymbol{S}^*(\boldsymbol{\theta}^v)$. To ensure stability, we control the learning rate by updating $\boldsymbol{\theta}^{v+1} = \boldsymbol{\theta}^v + \{\boldsymbol{M}^*(\boldsymbol{\theta}^v) + \boldsymbol{S}^*(\boldsymbol{\theta}^v)^T \boldsymbol{S}^*(\boldsymbol{\theta}^v)\}^{-1}\boldsymbol{S}^*(\boldsymbol{\theta}^v)$ in practice.

Note that the algorithm requires the specification of (K, J), which are typically unknown in practice. In this study, we propose to determine (K, J) using information criteria, the efficacy of which is evaluated in Section V-C.

## V. SIMULATION STUDY

This section assesses the proposed methods through simulation studies using synthetic data resembling the AURORA dataset. We generate data with $N$=200, $p$=23, $d$=3, $J = 3$, and $K = 3$. Each individual's number of observations, $T_i$, is uniformly sampled from $[50, 100]$. From this, we randomly select $T_i$ time points from $\{1, \cdots, 100\}$ to create sequences of $\delta_{it}$. Each individual's initial state is drawn from a multinomial distribution with probabilities $\boldsymbol{\pi} = (\frac{1}{3}, \frac{1}{3}, \frac{1}{3})$. Latent state trajectories are then generated based on transition probabilities $\boldsymbol{P}_{it}(\delta_{it})$, given individuals' features. Observation vectors $\boldsymbol{y}_{it}$ for an individual $i$ in state $j$ at time $t$ are then sampled from a normal distribution with mean $\boldsymbol{\mu}_j$ and covariance $\boldsymbol{\Lambda}_j \boldsymbol{\Lambda}_j' + \boldsymbol{\Psi}$, where $\boldsymbol{\Psi} = \boldsymbol{I}$. In the following, subsection V-A assesses model reliability by comparing empirical parameter estimates against true values; subsection V-B compares the effectiveness of our method against baseline methods; and subsection V-C explores the performance of information criteria in model selection.

### A. Simulation 1

To validate the estimation procedure, we implement the EM algorithm with true $J$ and $K$. Parameter initialization involves first fitting Gaussian Mixture Models to estimate groups, followed by EFM for each group. Guided by the insights from a pilot study, we set the maximum number of iterations for each replication at 100. The reliability and precision of the proposed methods are then evaluated from two perspectives: i) the accuracy of each individual parameter estimate and ii) the misclassification rate ($C_{mis}$), which quantifies the proportion of estimated states that diverge from the actual states.

The accuracy of parameters $\boldsymbol{\pi}$, $\boldsymbol{\mu}$, $\boldsymbol{\Lambda}$, and $\boldsymbol{\Psi}$ is assessed by calculating the average absolute difference (AAD) between the estimates and their true values, defined as $AAD(\boldsymbol{o}) = \frac{\sum_{i=1}^r |\hat{o}_i - o_i|}{r}$, $o_i$ is an individual entry in matrix $\boldsymbol{o}$ and $r$ is the total number of free parameters. The mean of AADs (standard errors in the parentheses) aggregated over 100 random seeds

| $\pi$ | $\mu$ | $\Lambda$ | $\Psi$ | $C_{mis}$ |
|---|---|---|---|---|
| .026 (.013) | .015 (.002) | .014 (.001) | .011 (.002) | .0024 (.0005) |

TABLE II
BIAS (STANDARD ERROR) OF THE PARAMETER ESTIMATES FOR EACH
TRANSITION MODEL PARAMETER $B_{jkl}$.

| $\boldsymbol{B}_{jk}$ | $B_{jk0}$ | $B_{jk1}$ | $B_{jk2}$ |
|---|---|---|---|
| $\boldsymbol{B}_{12}$ | .014(.142) | .017(.150) | -.080(.219) |
| $\boldsymbol{B}_{13}$ | -.021(.126) | .010(.125) | .016(.182) |
| $\boldsymbol{B}_{21}$ | -.018(.147) | -.004(.139) | .014(.214) |
| $\boldsymbol{B}_{23}$ | -.003(.112) | .004(.097) | .005(.200) |
| $\boldsymbol{B}_{31}$ | .003(.112) | .002(.107) | -.006(.192) |
| $\boldsymbol{B}_{32}$ | .006(.131) | -.032(.091) | .028(.232) |

are presented in Table I. These mean AADs for all parameter matrices are sufficiently close to zero with small standard errors, indicating effective parameter recovery. In Table II, we report the mean bias (standard error) for each parameter in the transition model, where the biases are all close to zero. Moreover, we present the mean (standard error) of $C_{mis}$ in Table I. On average, misclassification rates are only .24% (0.0005), highlighting the exceptional accuracy of the proposed EM algorithm in estimating latent states.

Intuitively, factors such as N, $T_i$, sizes of $J$ and $K$, variance $\Psi$, differences in $\mu_j$ and $\Lambda_j$ between states, and frequency of state transitions, can affect the performance of parameter estimation. Additional simulations in Appendix B reveals that estimation accuracy for $\mu$, $\Lambda$, $\Psi$, and $B$, as well as classification accuracy, improve when (i) common variances decrease, (ii) differences in $\mu_j$ and $\Lambda_j$ between states increase, (iii) $J$ decreases, or (iv) sample size ($N$) or panel length ($T_i$) increases. Increasing $K$ or using a $B$ that induces infrequent transitions slightly affects most parameters but enhances the precision of transition probability estimates, thereby reducing misclassification rates. Estimation of $\pi$ improves solely with increases in N or state-to-state differences in $\mu_j$.

*B. Simulation 2*

This section compares the performance of the proposed methods and the baseline approaches in correctly identifying latent states. Three benchmark methods are under our consideration: i) TM+independent HMM, which assumes independence among observed features given the states; ii) CFM+TM+HMM, which addresses interrelationships but inaccurately pre-specifies the latent factor structure by setting certain loading matrix entries to zero; and iii) EFM+HMM, which assumes a homogeneous transition probabilities matrix for all subjects. We first repeat the data generation process of Simulation 1. Then, we consider three additional scenarios by adjusting the state-to-state differences in $\mu_j$ to be closer ($\mu$: medium diff), increasing the similarity of the $\Lambda_j$ at different states ($\Lambda$: medium diff), and increasing the significance of the covariance matrix $\Psi$ ($\Psi = 2 \times I$), respectively.

As depicted in Figure 1, our proposed methods (HMFM) consistently outperform the benchmark methodologies in all settings. Regardless of sample size, our methods consistently achieve the lowest misclassification rate, nearly approximating zero, thereby emphasizing the importance of each component

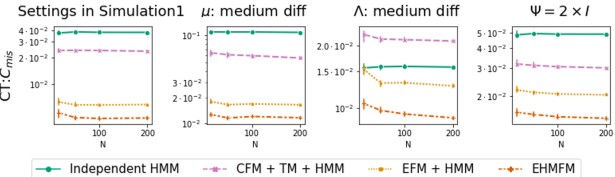

Fig. 1. $C_{mis}$ of various methods. The error bars represent the 95% CI. The first column shows the results under the settings we used in simulation 1. The last three columns summarize the results under different settings by varying the true value of $\mu$, $\Lambda$, and $\Psi$, respectively.

in our proposed models. Specifically, the comparison with TM+independent HMM shows the importance of accounting for the interrelationship between observed features; the comparison with CFM+TM+HMM reveals the risk of incorrectly specifying the interrelationship structure; and the comparison with EFM+HMM demonstrates the inadequacy of assuming homogeneous transition probabilities.

*C. Simulation 3*

Information criteria such as the Akaike information criteria (AIC) and the Bayesian information criteria (BIC) have been widely used in model selection [10], [30]. Within this simulation study, we investigate whether the AIC or BIC is reliable for determining J and K simultaneously. We repeat the data generation process of Simulation 1, but implement the proposed methods with a different set of $(J, K)$ for each replicate when fitting the generated data. Let $J = \{2, 3, 4\}$ and $K = \{2, 3, 4\}$. We consider all possible combinations of $J$ and $K$, yielding a total of nine fitted candidate models for each replicate. For 100 replications, both AIC and BIC consistently recommend the model with accurate $J$ and $K$. Therefore, we believe that the sample size and the number of observations per individual in the processed AURORA data will yield reliable information criteria-based model selection results and, consequently, reliable parameter estimation.

## VI. ANALYSIS OF THE AURORA DATA

Given the irregular data collection from mobile devices, we applied the CT-HMFM to smartwatch data from the AURORA study. Considering $J = 1, 2, \cdots, 6$ and $K = 1, 2, \cdots, 9$, we evaluate 54 candidate models. For each candidate model, the EM algorithm is implemented with multiple random seeds, and the seed yielding the highest estimated likelihood is selected. Finally, model comparison using AIC and BIC led to the selection of a model with $J = 3$ and $K = 8$. Subsequent subsections will discuss the interpretation of parameter estimates and biological insights from three perspectives: i) the interpretation of the three estimated states, ii) symptom co-occurrence patterns, and iii) the influence of demographic factors on transition probabilities.

*A. Interpretation of Hidden States*

To investigate the biological differences between states, we first focus on the selected features. Figure 2 depicts scaled sample means for each feature across states, with a 99% confidence interval (CI). Further pairwise Tukey tests indicate significant differences between states for nearly all features, except for amplitude, SWCK, L5, and NNskew.q3 between

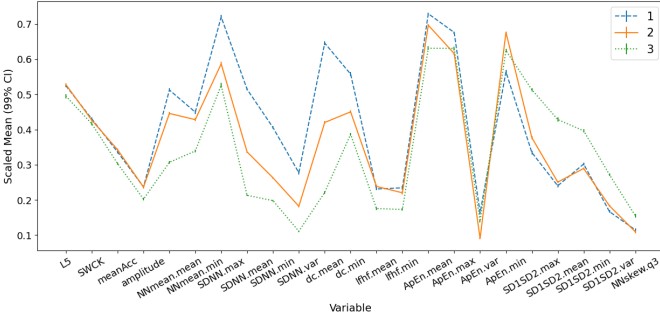

Fig. 2. Relative sample mean for features in each estimated state, with 99% CI error bars that are too small to distinguish.

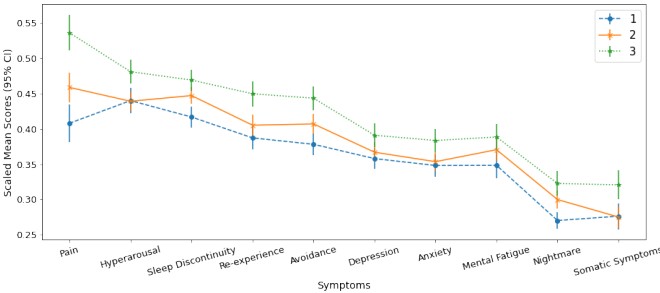

Fig. 3. Sample mean for each symptom in each estimated state. The error bars represent the 95% CI. 0 is the least severity, 1 is the greatest.

state 1 and state 2. Specifically, features related to average heart rate (NNmean), heart rate variability (SDNN), and heart deceleration capacity (dc) vary significantly across the three latent states, decreasing sequentially from state 1 to state 3. According to previous research, lower heart rate variability and deceleration capacity are associated with a higher mortality rate [23], [31], suggesting that states 1 through 3 represent decreasing levels of health, with state 1 being the healthiest. Additionally, activity levels are similar and higher in states 1 and 2 compared to state 3, indicating better overall health for participants in the first two states. Among features related to heart rate unpredictability (lfhf, ApEn, and SD1SD2), state 3 demonstrates significantly higher values for SD1SD2-related features but lower values for lfhf-related features compared to states 1 and 2, suggesting a different interpretation of the estimated states than our previous interpretation. However, it is important to note that the relationship between these features and the psychological or physiological state is neither straightforward nor unique [32].

To confirm the validity of the three states, we further compare their differences regarding self-reported symptoms from a flash survey. Based on the RDoC framework, ten latent constructs associated with APNS were developed using survey items selected by domain experts: Pain, Loss, Sleep Discontinuity, Nightmare, Somatic Symptoms, Mental Fatigue, Avoidance, Re-experience, and Anxious. Retaining only observations for each individual whose estimated states are known on the same day they submitted survey responses, we summarized the flash survey data with means and 95% CIs in Figure 3. Overall, state 1 exhibits the lowest severity level for all ten symptoms, while state 3 has the highest severity level. Tukey tests reveal no significant differences between states 1 and 2 in hyperarousal, re-experience, anxiety, and somatic

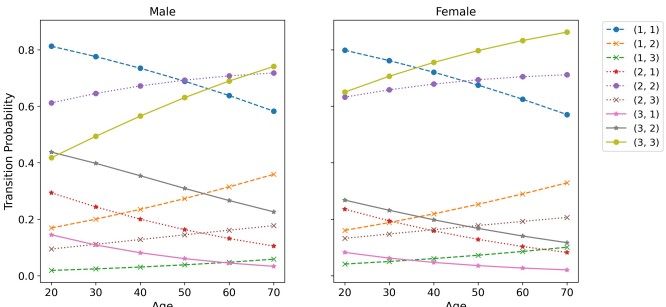

Fig. 4. Estimated transition probability. Fix $\delta_{it} = 1$. (a, b) indicates a transition from state a to state b.

symptoms, but both are significantly different from state 3 in these constructs. While the differences in nightmare and sleep discontinuity between states 3 and 2 are not significant, they are statistically more severe than in state 1. For mental fatigue and depression, only the difference between state 1 and state 3 is statistically significant. In summary, both the survey data and the AURORA data support our interpretation of the three latent states. State 1 is the healthiest, while state 3 indicates having the most severe APNS symptoms.

### B. Co-occurring Pattern of Symptoms

When studying the co-occurring of symptoms within each hidden state, we limit our attention to observations collected during the first week. For each state, the correlations between all ten symptoms are calculated. In state 1 (relative health state), there is a high degree of correlation (.782) between hyperarousal and anxiety, suggesting that patients in state 1 experiencing severe hyperarousal symptoms are also likely to suffer from severe anxiety. State 2 shows no highly correlated symptoms. In state 3 (the state with more severe disorders), symptoms such as depression, hyperarousal, anxiety, and re-experience are more likely to co-occur with pairwise correlation ranging from .716 to .91.

### C. Transition Probability

This section investigates the heterogeneity of 1-day transition probabilities among subjects, focusing on transitions with a time interval $\delta_{it} = 1$. We estimated the transition probabilities for males and females within the sample age range, as depicted in Figure 4. Lines with circles illustrate the probability of remaining in the same state, lines with stars indicate transitions to a more severe state, and lines with 'x' reflect the chance of improvement in psychological conditions.

Overall, both males and females have a tendency to remain in their current state, with infrequent state transitions, aligning with most literature. For males, the probability of staying at states 3 and 2 increases with age, while it decreases for state 1. Moreover, while the likelihood of psychological deterioration increases with age, the chance of improvement decreases. Specifically, while the probability of transitioning from the most severe state (state 3) to the healthiest state (state 1) approaches zero as age increases, the likelihood of the reverse transition increases as age decreases, with direct transitions between state 1 and state 3 being particularly rare. The female group exhibits a similar trend to the male group, but with

a higher likelihood of remaining in the most severe state (state 3) compared to males. In summary, our analysis of the AURORA data suggests that older patients are more likely to transition to more severe psychological states. Moreover, achieving psychological improvement becomes increasingly challenging as one ages.

## VII. Conclusions and Discussion

This paper investigates the unique challenges of analyzing longitudinal mobile health data, including interdependent variables with unknown interrelationship structures, heterogeneous transition probabilities, and irregular measurements. To address these issues, we propose a HMM-based model, the CT-HMFM, for multivariate longitudinal data collected irregularly. Furthermore, the performance of the corresponding Stabilized Expectation-Maximization algorithm for maximum likelihood estimation is supported by extensive simulation studies. Finally, we analyzed the AURORA data and drew biological findings comparable with previous research, implying that the mobile health data sourced from consumer-grade devices, together with the proposed methods, have the immense potential to facilitate mental health diagnostics and understand the dynamic transition mechanism.

The proposed methods can be extended in several ways. *First*, most entries in the estimated factor loading matrix are close to zero, indicating sparse factor loading matrices in real analysis. Although various methods (e.g., factor rotations and setting factor loadings below specific cutoffs to 0) are frequently used to simplify interpretation, the choice of these methods is subjective. The sparse exploratory factor loading analysis [33], [34] provides an automated approach to set the loading entries of redundant variables to 0, thereby enhancing the interpretability of loading matrices without reliance on subjective factors. Therefore, incorporating sparse regularization into the factor loading matrix is an important extension of our current work worth studying. *Second*, a large number of baseline covariates are typically available in real data. However, we have no prior knowledge about the significance of each covariate in determining the transition probability. Hence, integrating regularization into the transition model to assist with variable selection can be extremely useful. *Third*, the exceptional diversity of mental health makes it challenging to satisfy the key conditional independence assumption of the HMM, leading to potential violations and hence model bias [35]. This issue is exacerbated by the likely presence of autocorrelation among observations collected from the same subject. Therefore, adding a random effect to the current model to account for the inter-patient heterogeneity is a natural extension [10]. *Finally*, previous HRV-related studies are often conducted in well-controlled laboratory environments. Thus, all existing HRV feature extraction tools rely on resting-state heart rate data. However, heart rate data collected in open environments will inevitably contain additional noise. For example, it is reasonable to expect that HRV features corresponding to different activity states (e.g., exercising and resting) would differ significantly. Therefore, recognizing the

lack of tools to extract HRV features corresponding to different activity states, we believe it would be advantageous to develop a preprocessing pipeline to concurrently process heart rate and activity data to derive appropriate HRV features.

## Appendix A
### Technical Details

*A. Supplement for E-step*

Denote $\widetilde{\mathbf{\Lambda}}_j = (\mathbf{\Lambda}_j, \boldsymbol{\mu}_j) \in \mathcal{R}^{p \times (K+1)}$ and $\widetilde{\boldsymbol{z}}_{it} = (\boldsymbol{z}_{it}^T, 1)^T \in \mathcal{R}^{(K+1)}$. Each of the three parts has an explicit form:

$$h(\boldsymbol{\pi}) = \sum_{i=1}^{N} \sum_{j=1}^{J} \gamma_{ij}^v(1) log(\pi_j),$$

$$h(\{\boldsymbol{B}_{kj}\}_{k,j=1}^J) = \sum_{i=1}^{N} \sum_{t=2}^{T_i} \sum_{j,k=1}^{J} \epsilon_{ikj}^v(t) log(\boldsymbol{P}_{itkj}),$$

$$h(\boldsymbol{\Psi}, \{\mathbf{\Lambda}_j\}_{j=1}^J, \{\boldsymbol{\mu}_j\}_{j=1}^J) = \sum_{i=1}^{N} \sum_{t=1}^{T_i} \sum_{j=1}^{J} \gamma_{ij}^v(t) log |\boldsymbol{\Psi}|$$
$$+ \gamma_{ij}^v(t) \boldsymbol{y}_{it}' \boldsymbol{\Psi}^{-1} \boldsymbol{y}_{it} - 2\gamma_{ij}^v(t) \boldsymbol{y}_{it}' \boldsymbol{\Psi}^{-1} \widetilde{\mathbf{\Lambda}}_j E_{\boldsymbol{\lambda}^v}(\widetilde{\boldsymbol{z}}_{it} | \boldsymbol{y}_{it}, w_{it})$$
$$+ \gamma_{ij}^v(t) tr(\widetilde{\mathbf{\Lambda}}_j' \boldsymbol{\Psi}^{-1} \widetilde{\mathbf{\Lambda}}_j E_{\boldsymbol{\lambda}^v}(\widetilde{\boldsymbol{z}}_{it} \widetilde{\boldsymbol{z}}_{it}' | \boldsymbol{y}_{it}, w_{it})).$$

*B. Supplement for M-step*

By setting the first derivative of $h(\boldsymbol{\pi})$ to zero, the parameters related to the initial state distribution are estimated as: $\pi_j^{new} = \sum_{i=1}^{N} \gamma_{ij}^v(1) / \sum_{i=1}^{N} \sum_{k=1}^{J} \gamma_{ik}^v(1)$. Similarly, the parameters used to characterize the conditional distribution of $\boldsymbol{y}_{it}$ given $w_{it}$ are estimated by setting the first derivative of $h(\boldsymbol{\Psi}, \{\mathbf{\Lambda}_j\}_{j=1}^J, \{\boldsymbol{\mu}_j\}_{j=1}^J)$ equal to 0, with

$$\widetilde{\mathbf{\Lambda}}_j^{new} = \frac{\sum_{i=1}^{N} \sum_{t=1}^{T_i} \gamma_{ij}^v(t) \boldsymbol{y}_{it} E_{\boldsymbol{\lambda}^v}(\widetilde{\boldsymbol{z}}_{it} | \boldsymbol{y}_{it}, w_{it})'}{\sum_{i=1}^{N} \sum_{t=1}^{T_i} \gamma_{ij}^v(t) E_{\boldsymbol{\lambda}^v}(\widetilde{\boldsymbol{z}}_{it} \widetilde{\boldsymbol{z}}_{it}' | \boldsymbol{y}_{it}, w_{it})}.$$

Meanwhile, we got the updated estimation of $\boldsymbol{\Psi}$, $\boldsymbol{\Psi}^{new}$, equals

$$diag\Big\{ \sum_{i=1}^{N} \sum_{t=1}^{T_i} \sum_{j=1}^{J} \gamma_{ij}^v(t) \{\boldsymbol{y}_{it} - \widetilde{\mathbf{\Lambda}}_j^{new} E_{\boldsymbol{\lambda}^v}(\widetilde{\boldsymbol{z}}_{it})\} \boldsymbol{y}_{it}' \Big\} / \sum_{i=1}^{N} T_i.$$

## Appendix B
### Additional Simulation Results

This section presents additional simulation results that investigate the impact of various factors on estimation performance. Using the settings for simulation 1 in Section V-A as the baseline, we conducted eight additional sets of simulations, each varying one component while maintaining rest components as the baseline setup. These components include: i) sample size (N), ii) number of measurements for each individual ($T_i$), iii) J, iv) K, v) size of common variance $\boldsymbol{\Psi}$, vi) state-to-state difference in $\boldsymbol{\mu}_j$, vii) state-to-state difference in $\mathbf{\Lambda}_j$, and viii) transition frequency. For common variance $\boldsymbol{\Psi}$, we evaluated scenarios with $.1\boldsymbol{I}$, $.5\boldsymbol{I}$, and $1\boldsymbol{I}$ (baseline). For $\boldsymbol{\mu}_j$ and $\mathbf{\Lambda}_j$, we adjusted state-to-state differences in two additional settings. For the test evaluating the effect of transition frequency, a frequent transition is defined as the probability of remaining in the same state being less than 0.70. The $\boldsymbol{B}$ in the baseline setting corresponds to infrequent transition. For tests evaluating the effects of J, K,

TABLE III
THE MEAN (STANDARD ERROR) AADs OF $\pi$, $\mu$, $\Lambda$, $\Psi$, AND $B$, AND $C_{mis}$ OF ESTIMATIONS UNDER DIFFERENT CT SETTINGS.

| ADD | $\pi$ | $\mu$ | $\Lambda$ | $\Psi$ | $B$ | $C_{mis}$ |
|---|---|---|---|---|---|---|
| $\Psi = 1 * I$ | .026(.013) | .015(.002) | .014(.001) | .011(.002) | .120(.027) | .0024(.0005) |
| $\Psi = .5 * I$ | .026(.013) | .012(.002) | .011(.001) | .005(.001) | .119(.027) | .0003(.0001) |
| $\Psi = .1 * I$ | .026(.013) | .010(.003) | .007(.001) | .001(.000) | .119(.027) | .0000(.0000) |
| $\mu$: large diff | .026(.013) | .015(.002) | .014(.001) | .011(.002) | .120(.027) | .002(.0005) |
| $\mu$: medium diff | .027(.013) | .015(.001) | .015(.001) | .011(.002) | .124(.028) | .007(.001) |
| $\mu$: minor diff | .034(.017) | .017(.003) | .478(.057) | .011(.002) | .161(.040) | .085(.004) |
| $\Lambda$: large diff | .026(.013) | .015(.002) | .014(.001) | .011(.002) | .120(.027) | .002(.0005) |
| $\Lambda$: medium diff | .026(.013) | .014(.002) | .014(.001) | .011(.002) | .122(.029) | .006(.001) |
| $\Lambda$: minor diff | .026(.013) | .014(.002) | .014(.001) | .011(.002) | .122(.028) | .004(.001) |
| $B$: infreq transit | .026(.013) | .015(.002) | .014(.001) | .011(.002) | .120(.027) | .002(.0005) |
| $B$: freq transit | .027(.013) | .015(.002) | .014(.001) | .011(.002) | .137(.043) | .007(.001) |
| $J = 2$ | .027(.021) | .012(.002) | .012(.001) | .011(.002) | .087(.026) | .0015(.0003) |
| $J = 3$ | .026(.013) | .015(.002) | .014(.001) | .011(.002) | .120(.027) | .0024(.0005) |
| $J = 4$ | .024(.011) | .017(.002) | .016(.001) | .011(.002) | .150(.028) | .0032(.0005) |
| $K = 2$ | .026(.013) | .015(.002) | .014(.001) | .010(.002) | .124(.027) | .0069(.0008) |
| $K = 3$ | .026(.013) | .015(.002) | .014(.001) | .011(.002) | .120(.027) | .0024(.0005) |
| $K = 5$ | .026(.013) | .015(.002) | .015(.001) | .012(.002) | .119(.027) | .0005(.0002) |
| $N = 50$ | .052(.029) | .030(.004) | .028(.002) | .022(.003) | .254(.058) | .0027(.0008) |
| $N = 100$ | .042(.019) | .021(.003) | .020(.001) | .015(.002) | .175(.038) | .0025(.0006) |
| $N = 500$ | .018(.010) | .009(.001) | .009(.001) | .007(.001) | .076(.017) | .0024(.0003) |
| $10 \leq T_i \leq 30$ | .029(.014) | .029(.004) | .027(.002) | .021(.004) | .251(.061) | .0028(.0009) |
| $30 \leq T_i \leq 50$ | .027(.014) | .020(.003) | .019(.001) | .015(.003) | .180(.043) | .0025(.0005) |
| $100 \leq T_i \leq 150$ | .028(.015) | .012(.001) | .011(.001) | .008(.001) | .098(.021) | .0023(.0003) |

N, and $T_i$, the baseline setups are modified as indicated in Table III.

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
