# OpenReview forum: "Continuous-Time Hidden Markov Factor Model for Mobile Health Data: Application to Adverse Posttraumatic Neuropsychiatric Sequelae"
_IEEE.org/EMBS/BHI/2024/Conference — IEEE BHI'24_

### Official Review · Reviewer_F3cA · 2024-08-14
**Continuous-Time Hidden Markov Factor Model for Mobile Health Data: Application to Adverse Posttraumatic Neuropsychiatric Sequelae**

**Overall Rating:** 7
**Confidence:** 2

**Other Quality Metrics:**

(a) good
(b) excellent
(c) great
(d) good

**Questions For The Authors:**

Why do the patients are included specifically on Day 67? Is 50 days a special threshold for APNS patients?

Could you please provide more details about the AURORA dataset? What is the sampling frequency of signals? Does 258 subjects wear the same type of device?

**Strengths:**

By developing continuous-time discrete-state HMM, the irregular spacing of measurements, a common situation among wearable data, can be neglected.

**Summary Of The Paper:**

This paper develops a Continuous-Time Hidden Markov Factor Model (CT-HMFM) and a Stabilized Expectation-Maximization algorithm for parameter estimation to study Adverse Posttraumatic Neuropsychiatric Sequelae (APNS), a mental health disease common in veterans and trauma survivors, by leveraging objective mobile device data to improve understanding of these conditions. Traditional research methods rely heavily on subjective self-reports, leading to measurement errors and biases. The authors propose a novel approach using longitudinal data from mobile devices in AURORA study to identify and track homogeneous APNS states and their transitions over time.

**Weaknesses:**

The dataset section is not described clearly. Only the number of subjects is included in the description without the wearables data description.

---

### Official Review · Reviewer_kXb9 · 2024-08-14
**Review of Continuous-Time Hidden Markov Factor Model for Mobile Health Data: Application to Adverse Posttraumatic Neuropsychiatric Sequelae**

**Overall Rating:** 7
**Confidence:** 3

**Other Quality Metrics:**

Clarity of Writing: Good
Clinical Significance: Fair
Methodological Novelty: Excellent
Experiments and Results: Great

**Questions For The Authors:**

1. Can you please clarify if there are specific tools or software packages that you have developed or recommended for implementing the CT-HMFM model? Considering its complexity, how do practitioners or researchers apply this model in real-world settings?

2. How sensitive is the CT-HMFM model to the key assumptions, such as the independence of observations given latent states? Have you conducted any sensitivity analyses or explored extensions that might relax some of these assumptions?

3. Can you provide more details on the role of covariates in the transition model? How did you select the covariates used in your analysis, and what insights did you gain from them? Are there any covariates you found particularly influential or surprising in their impact?

**Strengths:**

The CT-HMF model addresses mobile health data's challenges, such as irregular measurement intervals and complex interdependencies among variables. The model's ability to handle unevenly spaced measurements and interdependent variables makes it well-suited for analyzing longitudinal mobile health data, which often suffers from these issues. This capability is critical for accurately modeling real-world data collected in uncontrolled environments. The CT-HMFM allows transition probabilities to vary based on covariates, capturing heterogeneous transition patterns across individuals. This aspect of the model is crucial for understanding individual-level dynamics in health conditions, enabling more personalized insights and potentially more tailored interventions. The extensive simulation studies in the paper demonstrate the model's robustness and accuracy in parameter estimation, latent state identification, and overall performance. This thorough validation builds confidence in the model's applicability to real-world data. The successful application of the model to data from the AURORA study, particularly in analyzing heart rate variability (HRV) and activity data, showcases the model's practical utility and potential to uncover meaningful insights into trauma recovery and mental health.

**Summary Of The Paper:**

The paper presents a Continuous-Time Hidden Markov Factor Model (CT-HMFM) for analyzing longitudinal mobile health data. This model addresses the challenges of interdependent variables and irregularly spaced measurements. It combines Continuous-Time Hidden Markov Models (CTHMM) with Exploratory Factor Analysis to identify latent states and capture transitions between these states as a function of covariates in a continuous-time framework. The paper also describes an Expectation-Maximization (EM) algorithm for parameter estimation. Extensive simulation studies validate the accuracy and performance of the model, showing that it outperforms benchmark models. The CT-HMFM is applied to data from the AURORA study, focusing on heart rate variability (HRV) and activity data. It provides insights into the dynamic transitions among different health states in trauma survivors and the influence of demographic factors on these transitions. The model is presented as a robust tool for studying Adverse Posttraumatic Neuropsychiatric Sequelae (APNS) and has the potential for broader applications in mental health research.

**Weaknesses:**

The model's complexity, especially integrating CTHMM with Exploratory Factor Analysis, may make it challenging to apply in real-world settings outside of academic research. Practitioners who could benefit from this model might find it challenging to implement due to the advanced mathematical and computational requirements. The model relies on several assumptions, such as the independence of observations given the latent states and the structure of the factor model. If these assumptions are violated in practice, the model's performance could be compromised. The paper mentions the inclusion of covariates in the transition model, but the exploration of which covariates are most influential or how they impact the results is limited.

---

### Official Review · Reviewer_tmtE · 2024-08-27
**Innovative improvement on HMM model applied on PNAS**

**Overall Rating:** 8
**Confidence:** 3

**Other Quality Metrics:**

(a) Clarity of writing: great
(b) Clinical Significance: excellent
(c) Methodological Novelty: excellent
(d) Experiments and Results: good

**Questions For The Authors:**

I am confused about why the authors used simulation studies to prove the model's usefulness in classification but did not conduct the same classification task on the AURORA dataset. I may have misunderstood something, but the authors' clarification would greatly help.

**Strengths:**

1. Intuitive and innovative model idea.
2. Very indepth and clear description of the model.
3. Strong simulation study to verify the model's usefulness.
4. Proofs shown in the appendix.
5. Model is highly interpretable, capable of generating great insights.

**Summary Of The Paper:**

In this paper, the authors proposed using a new method of HMM, named CTHMFM, for analysis of Adverse posttraumatic neuropsychiatric sequelae (APNS). The model is improved upon the base HMM with the addition of the State-Specific Measurement Model (which models the correlations among variables) and the a transition model (TM) to capture heterogeneous transition patterns. This CTHMFM is careful described and verified using simulated tasks, and then subsequently applied on the AURORA dataset. The resulting interpretable model is used to generate clinical insights.

**Weaknesses:**

1. Lack of a code repo for the model, which could have other strong use cases.
2. Lack of a direct classification verification for PNAS.

---

### Decision · Program_Chairs · 2024-09-23

Accept